# Towards Reference Values for Malondialdehyde on Exhaled Breath Condensate: A Systematic Literature Review and Meta-Analysis

**DOI:** 10.3390/toxics10050258

**Published:** 2022-05-18

**Authors:** Veronica Turcu, Pascal Wild, Maud Hemmendinger, Jean-Jacques Sauvain, Enrico Bergamaschi, Nancy B. Hopf, Irina Guseva Canu

**Affiliations:** 1Center for Primary Care and Public Health (Unisanté), University of Lausanne, Route de la Corniche 2, 1066 Epalinges, Switzerland; veronica.turcu@unisante.ch (V.T.); pascal.wild@unisante.ch (P.W.); maud.hemmindinger@unisante.ch (M.H.); jean-jacques.sauvain@unisante.ch (J.-J.S.); nancy.hopf@unisante.ch (N.B.H.); 2Department of Public Health and Pediatrics, University of Turin, Via Zuretti 29, 10125 Turin, Italy; enrico.bergamaschi@unito.it

**Keywords:** malondialdehyde, exhaled breath condensate, meta-analysis, reference values

## Abstract

Many pathological conditions and certain airway exposures are associated with oxidative stress (OS). Malondialdehyde (MDA) is an end-product of the oxidation of lipids in our cells and is present in all biological matrices including exhaled breath condensate (EBC). To use MDA as a biomarker of OS in EBC, a reference interval should be defined. Thus, we sought to summarize reference values reported in healthy adult populations by performing a systematic review and meta-analysis using a standardized protocol registered in PROSPERO (CRD42020146623). Articles were retrieved from four major databases and 25 studies with 28 subgroups were included. Defining the distribution of MDA measured in reference populations with a detection combined with a separation technique still represents a challenge due to the low number of studies available, different analytical methods used, and questionable methodological qualities of many studies. The most salient methodological drawbacks have been in data collection and reporting of methods and study results by the researchers. The lack of compliance with the recommendations of the European Respiratory Society and American Thoracic Society was the major limitation in the current research involving EBC. Consequently, we were unable to establish a reference interval for MDA in EBC.

## 1. Introduction

Oxidative stress is an imbalance between production and elimination of reactive oxygen species (ROS), and represents a common pathological pathway for many inflammatory diseases, including chronic and acute lung conditions, such as chronic obstructive disease and asthma [1]. The biological effects of excess ROS production are their interactions with membrane lipids, a process known as lipid peroxidation, which yields a series of secondary molecules capable of exacerbating the oxidative damage [2]. Malondialdehyde (MDA) is one of the end-products of lipid peroxidation and has been largely investigated as a biomarker of oxidative stress. MDA correlates with different pathological states, such as cardiovascular disease and atherosclerosis [3,4], diabetic disease [5], Alzheimer’s, DNA damage, and ageing [6]. MDA has also been extensively investigated as a biomarker of pulmonary oxidative stress [7] and in evaluating the impact of air pollution, smoking [8], or respiratory exposure in occupational environments such as exposure to heavy metals [9] and particles [10].

MDA has attracted attention and interest as a biomarker of oxidative stress, because it can be detected in a variety of biological matrices, i.e., blood, urine, and exhaled breath condensate (EBC), and has shown good correlations with other biomarkers of lipid peroxidation [11]. 

EBC, the biological matrix considered in this meta-analysis, can be obtained by cooling an individual’s exhaled air using a noninvasive and safe procedure. This condensate contains low-volatile compounds and aerosolized airway lining fluid [12] which have been diluted by water [13]. According to current recommendations from the joint task force of American Thoracic Society and European Respiratory Society, EBC collection can be performed by asking an individual to breathe normally into a tube connected to a cooling apparatus while wearing a nose clip for a period of time until a reasonable amount of condensate is collected [14]. The current ERS technical standards also include specific recommendations as follows: (a) define and report EBC volume, time of collection, collection temperature, breathing pattern, prevention of salivary and environmental contamination, ambient conditions; (b) keep the time between collection of EBC and analysis as short as possible; (c) report results in terms of concentration per 100l of EBC or per minute, in order to reduce the confounding factor of the dilution of substances in the EBC; (d) report data concerning the intra- and inter-assay variability, limit of detection, sensitivity of detection methods, and more generally describe the methods and techniques used to collect and analyze the EBC; (e) note and report the time between food intake and EBC collection; (f) schedule collection for the same time of day.

The same task force, in 2005, also released a first series of methodological recommendations which were more generic than the 2017 recommendations, and the use of a nose clip and a saliva trap have been formulated since the earlier version. The biomarkers in EBC originate from different regions of the respiratory tract, from the lower airways and lungs up to the oropharynx and nasopharynx [15]. Hence, the use of a nose clip is recommended to prevent collection of mediators formed in the nose and the sinuses and to exclusively collect from the lower airways, as well as to ensure that no sample is lost through nasal respiration [16]. Similarly, saliva contamination needs to be avoided since it may contain the same mediators of interests but derived from other anatomical sites than the lower respiratory zone. It has been proposed that a saliva trap is sufficient, since there is a need for a sensitive test for salivary alpha-amylase. 

Other recommendation in the 2005 version included performing the analysis as soon as possible after collection, noting and reporting temperature collection, and detailing the collection devices if custom materials are used especially for the contact surface. Although a collection time of 10 min has been recommended, further research on the effect of collection time and temperature versus expiratory volume is needed [17]. Concerning the collection and analysis methods, the 2005 ATS/ERS recommendations were to provide sufficient details to ensure the reproducibility of technique. 

Furthermore, reporting participants’ details, such as smoking habits, race, age, sex, as well as eating and drinking behaviors have also been recommended since the first version of the task force [14]. We found it crucial to detail these two series of recommendations from the joint task force of the American Thoracic Society and European Respiratory Society, as this could have affected the different study protocols in the studies reviewed for this meta-analysis. 

MDA concentrations in EBC are usually quantified after collection and storage at low temperatures for up to 6 months, by using different analytical and separation methods: high performance liquid chromatography (HPLC) coupled with mass spectrometry (MS) or UV photometry detection. MDA is derivatized with either thiobarbitoric acid (TBA) or 2,4-dinitrophenylhydrazine (DNPH), with TBA being less specific for MDA [18]. 

Different levels of MDA in the EBC have been detected in some physiological or pathological conditions characterized by oxidative stress, but the absence of established or recognized reference intervals has limited the interpretation of the findings of the MDA as a biomarker in different settings. Thus, the current research has as a primary objective to investigate the possibility of defining reference intervals for MDA in EBC.

## 2. Materials and Methods

The present study focuses on MDA measured in EBC and is part of a series of systematic reviews and meta-analyses for several oxidative stress biomarkers in two different matrices: urine and EBC [19]. 

Our protocol followed recommendations from the Preferred Reporting Items for Systematic Reviews and Meta-Analysis (PRISMA) [20] and is registered with the International Prospective Register of Systematic Reviews (PROSPERO, registration number CRD 42020146623) [21].

### 2.1. Literature Research

Research for scientific literature was conducted on four different databases (The Cochrane Central Register, EMBASE, PubMed, and Web of Science), from journal apparition up to October 2020. We performed a quick search for the period covering the end of 2020 to end of 2021 to confirm that no new data had been published that could change our outcome. The research string was constructed in collaboration with the library of Unisanté and their documentarist using a combination of Medical Subject Headings (MeSH) terms and key words. The complete research string has been described elsewhere (www.doi.org/10.16909/dataset/17, accessed on 15 February 2022) and is available at the Unisanté data repository. 

### 2.2. Study Selection

We included only original research studies written in English that reported MDA concentrations in EBC among human healthy adults. The flowchart in Figure 1 represents the selection process of the studies that were ultimately included in our meta-analysis. First, we identified and excluded duplicates, and second, we used the Rayyan software [22] for title and abstract screening. 

The exclusion criteria were: MDA measured in any other matrix than EBC (e.g., blood/plasma and sperm) in vitro or non-human studies (animal or plant), incorrect MDA abbreviation (e.g., mass drug administration and methylenedioxyamphetamine), and inappropriate types of publications such as reviews or conference papers. In accordance with our registered protocol, our research string was built to cover an entire series of OS biomarkers, and was not limited to MDA. Consequently, our initial pool of articles contained articles that did not refer to MDA. Another factor that made this pool large was that authors often chose keywords that were very generic such as “oxidative stress” and “biomarkers of oxidative stress”. 

### 2.3. Data Extraction

We extracted data according to a previously published protocol [19], including a list of parameters regarding technical aspects of the analytical methods used to measure MDA in EBC and lifestyle factors such as body mass index (BMI) and vitamin supplement intake. We only extracted data on healthy subjects who were free of any disease, especially inflammatory conditions, and who were free of known inhalation exposure, or, where possible, who were healthy subjects before an exposure, i.e., baseline data and, if possible, all subgroup-specific data. 

### 2.4. Quality Assessment

Articles that underwent data extraction were also evaluated using a standardized quality assessment checklist, previously used in other studies [22]. This allowed us to appraise four different domains: study design and risk of bias, technical and analytical methods, study sample, and data interpretation. We assessed different criteria for each domain and graded them with scores from 1 to 3. The output was an overall score ranging from 0 to 27. In this range, two cut-offs and three quality categories were established: “low” (scores less than or equal to ≤9), “moderate” (scores less than or equal to ≤18), and “high” (scores greater than 19). The checklist used to perform the quality assessment is detailed in Appendix A and the overall scores of articles elected for data extraction are listed in Table 1.

### 2.5. Statistical Analysis

We computed geometric means (GMs) and geometric standard deviations (GSDs) as the basis of the meta-analysis. For value conversion and unit harmonization, we used different formulae according to types of data available and as previously described in [63]. 

Regarding the meta-analysis itself, the first step involved estimating the geometric means (GMs), and then representing the data with forest plots since they illustrated the heterogeneity and the overall combined results from individual studies. Between-study heterogeneity was evaluated with a Q test and displayed for each category in the forest plots. Finally, we modeled the Log-transformed GMs using linear models. We reported the results in ng/mL of MDA in EBC. 

## 3. Results

Among the 40 full-text articles from which we extracted the data, several studies had to be excluded due to statistical errors such as lack of variability estimates or impossible reported ranges for the studied groups. We also excluded from the meta-analysis several study groups that demonstrated a coefficient of variation (CV), either <20% or >200%. The former corresponds to the analytical variability alone, and implies no between-subject variability. The latter implies that the study is uninformative. One study was excluded as the results had been reported using inappropriate units. The whole selection process is illustrated in Figure 1. 

Since different factors were indicated as potential confounding factors in the literature, we wanted to assess their influence. Therefore, the results were stratified by gender, age, smoking status, EBC collection device, duration of EBC collection, use of nose clip, and analytical methods combined with their detection methods. We created a separate group identified by the abbreviation NA, meaning “not available”, which was assigned when the information was not given in the article. This resulted in the description of 28 study subgroups. Figure 2 shows the boxplots concerning the collection device (Figure 2a), duration of collection (Figure 2b), and gender groups (Figure 2c).

Among the included articles, we identified the following analytical and separation methods combined with a derivatizing agent: high-performance liquid chromatography/thiobarbituric acid HPLC/TBA, spectrophotometry/TBA, HPLC—mass spectroscopy, HPLC-MS/initrophenylhydrazine (DNPH), and HPLC without mentioning the derivatizing agent and detection technique. A mixed effect regression analysis was performed to verify differences among the MDA GMs measured by different analytical methods. The same was done concerning the use of a nose clip versus no use of a nose clip, age categories, and among smokers and non-smokers. For each category, a factor-specific Wald test within the model was equally performed. Table 2 shows the results of the mixed-effects regression analysis. As compared with MDA GMs measured in EBC by HPLC/TBA, which was considered to be the reference method, the MDA GMs measured by MS/DNPH were significantly lower. The GMs of EBC MDA collected in uncertain conditions were higher as compared with the GMs of EBC collected while wearing a nose clip, which was the reference group. GMs of MDA in EBC higher than the reference were also found in smokers and in subjects 50+ years old as compared with in subjects younger than 30 years old.

These results were equally presented as separate forest plots, which better illustrate the heterogeneity for the age categories (Figure 3) and smoking habits (Figure 4).

The predicted GMs in ng/mL for MDA measured in EBC is provided in Table 3, as well as the 95% confidence interval for each parameter considered. We observe that most of our predicted values fall in the interval of 0.18–2.87 ng/mL. 

## 4. Discussion

This meta-analysis showed that subjects older than 50 years had different MDA in EBC values from those younger than 30 years of age. Hence, for healthy subjects aged 30–50 years, we observed MDA in EBC values ranging from 0.18 to 2.87 ng/mL. Another observation we made was that smokers had higher values, almost double, of MDA in EBC as compared with non-smokers.

These findings are in line with recent studies that have investigated the link between aging and oxidative stress and have demonstrated the age dependence of oxidative stress in humans, although on different matrices. When measuring several biomarkers of oxidative stress, Pinchuk et al. revealed that MDA increased with age, and especially in post-menopausal women [63], and they attributed this to changes in estrogen levels. A meta-analysis regarding urinary MDA has made similar findings, that is, a trend of MDA values increasing with age, probably through the pathway of proteasome dysfunction. In addition, the association between smoking and increased oxidative stress has been well established, i.e., smoking induces the production of reactive oxygen species (ROS) that may activate a cascade and generate further molecular damage [64]. Although limited, there is evidence that MDA also increases in the EBC of smokers and not just in biological matrices such as the blood [65]. 

Our meta-analysis revealed significant variations in MDA levels according to the use of a nose clip during the EBC collection. The interpretation of such a finding is flawed by the inclusion of studies that did not specify if they used the nose clip or not. The finding itself would be coherent with the present recommendations to use a nose clip.

Nevertheless, we do not suggest the above-mentioned value intervals as reference intervals, due to several factors, which are discussed in the following paragraphs, such as the high heterogeneity, the low number of studies, or the use of proximity samples in the majority of the selected studies.

### 4.1. Heterogeneity

Our meta-analysis had a high level of heterogeneity, which we failed to reduce in subgroup analysis. This may be due to several factors, such as the heterogeneity in several steps: sample collection, chemical analytical analysis, data collection, and data reporting. Missing information regarding these steps also led to mitigated quality scores, mostly being moderate to low (Table 1). 

#### 4.1.1. Heterogeneity Related to the Collection of EBC

We checked the compliance with current methodological recommendations for EBC collection according to the date of article publication, and found three studies performed after 2017; two of these studies respected the guidelines concerning the EBC collection. The majority of the 26 studies included in the qualitative analysis was published between 2005 and 2017. Even though these studies should have adhered to the ERS/ATS 2005 recommendations, ten studies stated that they did not use a nose clip or did not provide the information regarding the use of a nose clip. The reasons for omitting this information or for not providing a nose clip to the participants as recommended by the guidelines remain to be unclear. Similar observations were made concerning the EBC devices, collection duration, and time of day the EBC collection was performed. Further, for about a fourth of the studies, we did not have enough information from the published article to allow us to understand if saliva contamination had been excluded. 

#### 4.1.2. Heterogeneity Related to the Analytical Method

Studies in our meta-analysis reported the use of different MDA derivation methods. The authors presented improved or modified already existing methodologies, which probably contributed to the overall heterogeneity. Moreover, data concerning the limit of detection and/or the limit of quantification, exclusion of potential contamination, and parameters of transportation or storage were partially revealed in the articles.

#### 4.1.3. Heterogeneity Related to Data Collection and Reporting

Concerning the data collection and reporting, almost 20% of the studies did not specify the distribution of males and females among their control participants, 25% of the studies did not clearly report the smoking status, and 60% of the studies did not provide information on the BMI. Ideally, in addition to these basic data, more information concerning lifestyle factors and dietary information, such as vitamin intake, medical regime, occasional drug use, or living in an area with traffic-related heavy air pollution, should be taken into consideration, to better study their influence on MDA levels [66]. As these factors might influence the level of MDA in a reference sample, MDA reference intervals should be stratified accordingly, avoiding any misleading estimation. 

## 5. Study Strengths and Limitations

The strengths of our meta-analysis are our rigorous research protocol, the exhaustive literature search by a documentarist using four databases, and our effort to harmonize units and generate background values as geometric means. Given the time lapse between literature research and data extraction, we also checked for studies published after October 2020, but this did not change our dataset.

Nevertheless, in this meta-analysis, the reduced number of performed studies and their quality represented major limitations. Ultimately, this also led to the high overall heterogeneity, which we did not manage to reduce even by performing a subgroup analysis. 

## 6. Recommendations

Good quality and complete datasets are required to derive a reference interval of MDA in the EBC; however, the heterogeneity among studies observed here, prompts us to strongly recommend that researchers collect and report complete data information in future studies. This pertains especially to: (a)Collection and reporting of demographics and health status, namely gender, age, smoking status, BMI, diet, living area, sports, and respiratory functions to insure that the study sample does really correspond to the reference sample, and that it is representative of the general healthy population and all its subgroups.(b)Collection and reporting of the time of EBC collection and volume of collected EBC as this is one ERS/ATS recommendation that has not yet been fulfilled. We strongly recommend avoiding the use of non-validated EBC collection devices since it has been shown that the inner surface of the collection equipment can interfere with the determination of biomarkers [67], presumably, also with MDA determination. Therefore, EBC collections need to be standardized. A greater number of studies respecting the current guidelines would help to get closer to establishing a standardized EBC collection method.(c)Reporting of analytical methods that may affect MDA quantification and other factors influencing or supposed to influence the results, related to the analysis itself or collection device and material.

## 7. Conclusions

Our systematic review and meta-analysis could not provide any reference interval for MDA in EBC, due to the large heterogeneity among studies and the methodological limitations encountered. Further research is needed to, first, understand the influence of demographic and collection parameters on the MDA values in EBC, and secondly, to harmonize analytical methods. This and a sufficient number of high-quality studies would help to define MDA reference values in EBC.

## Figures and Tables

**Figure 1 toxics-10-00258-f001:**
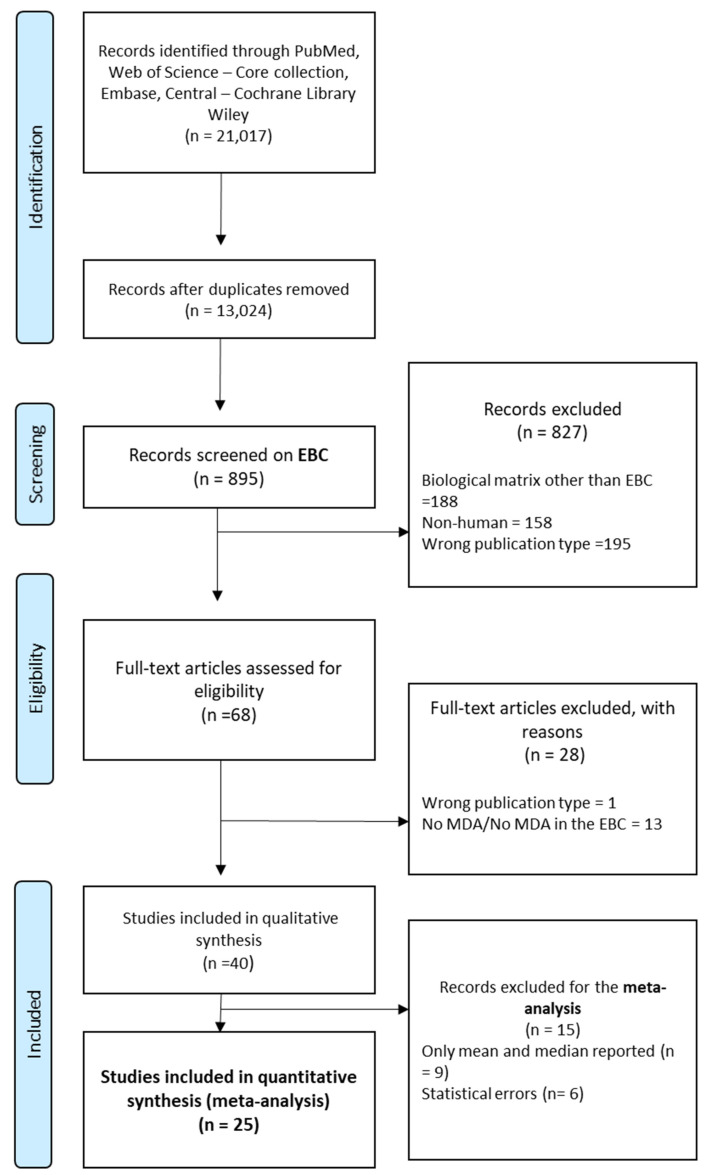
Flowchart of the article selection process.

**Figure 2 toxics-10-00258-f002:**
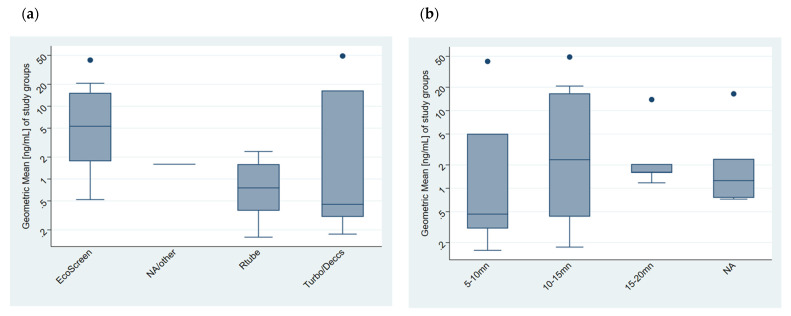
MDA concentration in EBC stratified according to: (**a**) Collection device; (**b**) duration of collection (minutes); (**c**) gender group (no papers with females only). All presented results exclude the outliers and units of the GMs are ng/mL.

**Figure 3 toxics-10-00258-f003:**
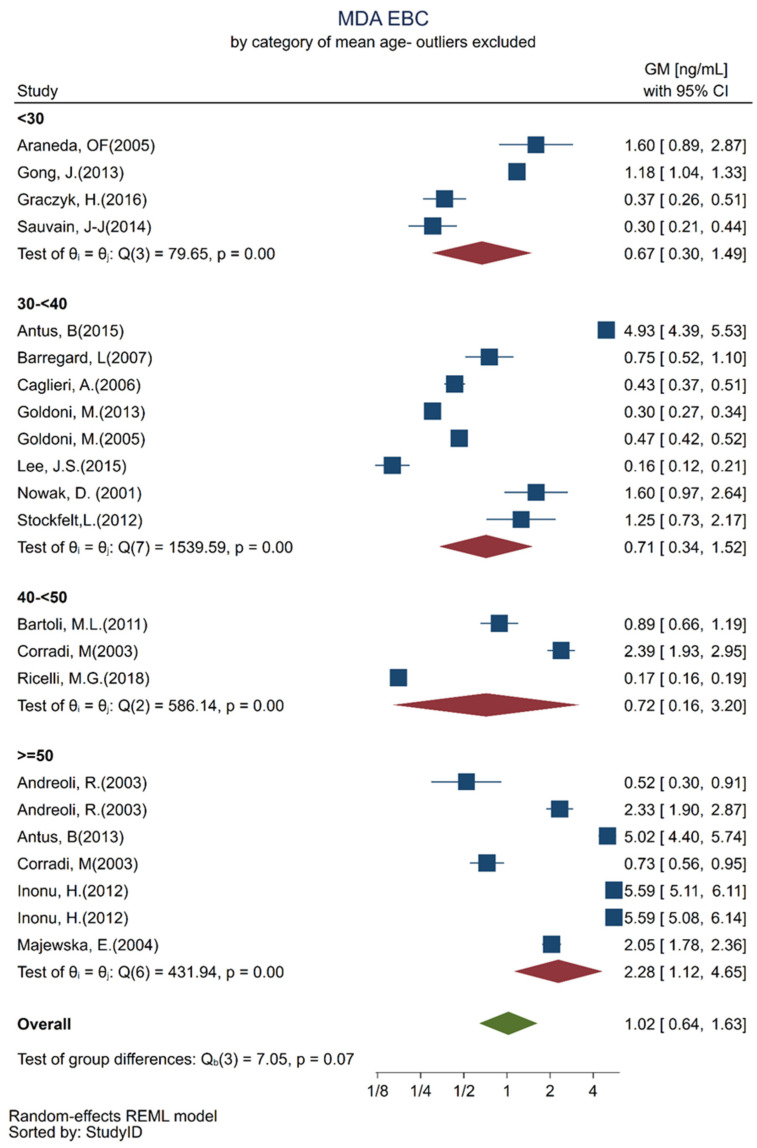
Forest plots of MDA in EBC concentration quantified in healthy adult participants by age group: population with a mean age less than 30 years [28,43,46,58]; between 30 and 40 years [25,31,34,41,42,49,51,59], between 40 and 50 years [32,36,55] and above 50 years [24,26,36,46,50].

**Figure 4 toxics-10-00258-f004:**
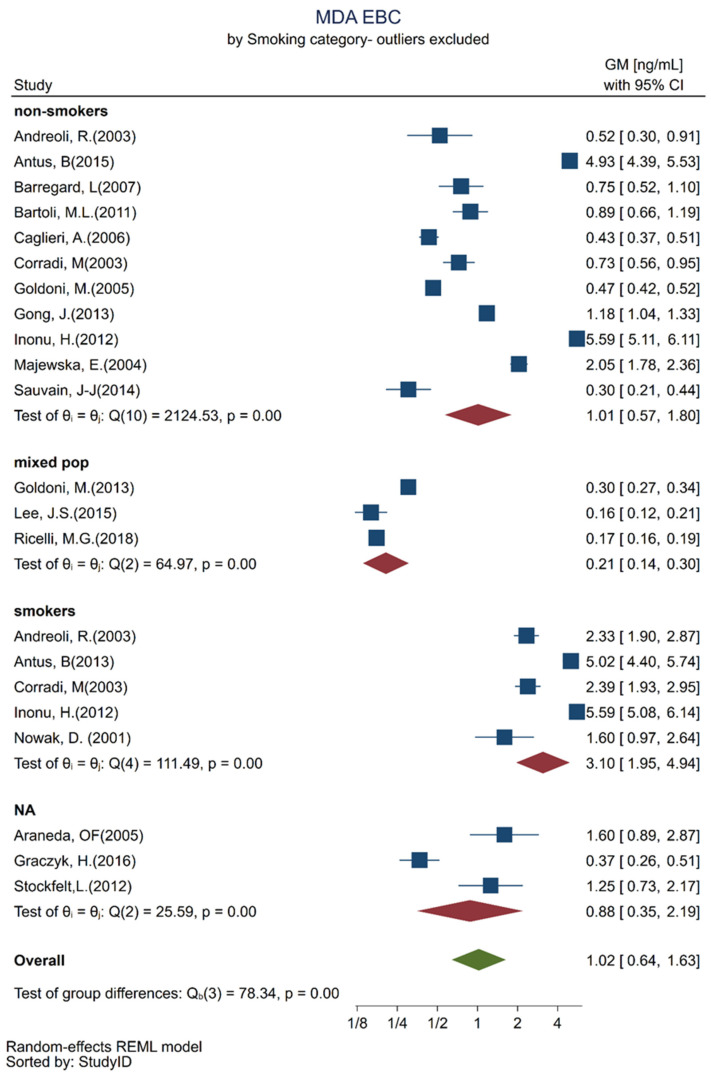
Forest plots of MDA in EBC concentration quantified in healthy adult participants considering their smoking habits: non-smokers exclusively [24,25,31,32,34,36,42,43,46,50,58]; mixed population of smokers and non-smokers [41,49,55], smokers exclusively [24,26,36,46,51] and unknown smoking status [28,44,59].

**Table 1 toxics-10-00258-t001:** Qualitative synthesis of articles that underwent data extraction.

Author, Year, Country	Study Objectives	Population Studied and Number of Participants	Control Population	Method of EBC Collection and Analysis	Main Findings	Quality Score
Aksu, 2012, Turkey [23]	To investigate lower airway inflammation using malondialdehyde and total protein measurement in exhaled breath condensate	12 Mild asthma, 53 persistent rhinitis, and 16 concomitant asthma and rhinitis	13 Control subjects (5 males and 8 females)	EBC was collected with a homemade apparatus for max 15 min at 10.00 am, no nose clip, immediately stored at −80 °C	MDA and total protein levels in EBC did not differ among the groups	14 Moderate
Andreoli, 2003, Italy [24]	Method development	None	2 Non-smoking and 13 asymptomatic smoking control subjects	EBC collected on EcoScreen, 15 min with nose clip, and then quickly stored at −80 °C	MDA concentrations were consistent with those determined in EBC using an independent HPLC-FLD method, mass spectrometry to EBC analysis could play a fundamental role in the validation of EBC as a new accessible biological matrix and in the definition of standardized EBC collection	12 Moderate
Antus, 2015, Hungary [25]	To investigate oxidative stress in cystic fibrosis (CF) patients	40 Patients with CF	25 Control subjects (13 males and 12 females)	EBC collection with an EcoScreen condenser using MDA reagent kit, MDA concentrations in EBC were measured with an isocratic HPLC system		19 High
Antus, 2013, Hungary [26]	To investigate MDA levels in patients with acute COPD exacerbation	55 Patients with COPD	20 Healthy controls, ex-smokers (10 females and 10 males)	EBC collected for 10 min EcoScreen condenser, samples were stored frozen at −80 °C before analysis, isocratic HPLC system using MDA reagent kit		12 Moderate
Antczak, 1997, Poland [27]	To determine whether asthmatic patients exhale more H_2_O_2_ and TBARS than healthy subjects	21 Asthmatic patients	10 Healthy subjects (6 males and 4 females)	EBC collected in a tube installed in a polystyrene foam container filled with ice and salt, for 20 min, and aliquotes were transferred in Eppendorf tubes stored at −80 °C for no longer than 7 days until measurement, TBAR species in expired breath condensate was determined according to the method of YAGI	H_2_O_2_ was 26 times higher in asthmatic EBC as compared with TBARS, these findings were statistically significant	15 Moderate
Araneda, 2005, Chile [28]	To investigate lung oxidative stress in subjects exercising in moderate and high altitude	None	6 Healthy males	EBC collected with self-constructed device for 20 min, condensated at −5 °C, and stored in liquid nitrogen, MDA was determined with HPLC according to Larstad	Lung OS may constitute a pathogenic factor in acute mountain sickness	11 Moderate
Araneda, 2012, Chile [29]	To investigate the impact of an endurance race on pulmonary pro-oxidative formation and lipoperoxidation	41 Healthy recreational runners divided into 3 groups according to the races they decided to run (10, 21.1, and 41.2 km) non-smokers	Two groups (10 and 42.2 km) with MDA in EBC, pre-race data was available just for these subjects	EBC collected with self-constructed device for 10 to 20 min, with nose clip, condensated at −5 °C, and stored in liquid nitrogen, MDA was determined with HPLC according to Larstad	Intense prolonged exercise favors an increase in pulmonary pro-oxidative levels, with no modifications on lipoperoxidation; Running time relates to the magnitude of acute post exercise pro-oxidative formation	12 Moderate
Araneda, 2013, Chile [30]	To investigate pro-oxidative, lipid peroxidation, and inflammation in EBC, in subjects running 10 km	None	10 Physically active healthy subjects (9 males and 1 female), EBC measured 20 min before the race	EBC collected with self-constructed device for 10 to 15 min, with nose clip, condensated at −5 °C, and stored in liquid nitrogen, MDA was determined with HPLC according to Larstad	Unlike the previous results obtained in amateur runners, in physically active subjects, 10 km produces an increase in oxygen- and nitrogen-derived pro-oxidative species, without early peroxidation	7 Low
Barregard, 2007, Sweden [31]	To examine whether short-term exposure to wood smoke in healthy subjects affects markers of pulmonary inflammation and oxidative stress		13 Healty subjects (6 males and 7 females), EBC measured prior to exposure	EBC was collected with an Rtube, with nose clip, until 80 L were collected, samples were frozen at −20 °C, MDA was determined with HPLC according to Larstad	Wood smoke exposure affects the respiratory tract, especially the lower airways	10 Moderate
Bartoli, 2011, Italy [32]	To assess the usefulness of MDA in EBC in different groups of pulmonary diseases	64 Subjects with asthma, 19 with bronchiectasies, 73 with COPD, 38 with idiopathic pulmonary fibrosis	14 Healthy non-smoking subjects	EBC was collected on an EcoScreen device, during 15 min, samples were immediately stored at −80 °C and analyzed within 6 months, with HPLC according to Larstad	Subjects with chronic airway disorders have increased levels of MDA in EBC, MDA concentrations in EBC are related to FEV1 and neutrophilic inflammation, particularly in COPD patients.	15 Moderate
Brand, 2013, Germany [33]	To investigate if short-term exposure to welding fumes results in changes in lung function and early stages of inflammatory reactions		12 Healthy non-smoking males	EBC was collecte on an ECOSCREEN device wearing a nose clip, for 20 min, samples stored at −80 °C until analysis, MDA was derivatized with DNPH, and then separated by HPLC and determined by mass spectrometry	In healthy, young subjects, neither changes in spirometry nor changes in inflammatory markers measured in exhaled breath condensate could be detected after short-term exposure	12 Moderate
Caglieri, 2006, Italy [34]	To investigate chromium levels in EBC of workers exposed to Cr(VI) and to assess their relationship with biochemical changes in the airways by analyzing EBC biomarkers of oxidative stress	24 Chrome-plating workers	25 Control subjects (13 males and 12 females, 5 ex-smokers and 20 non-smokers)	EBC was collected on TURBO DECCS, condensation temperature of −5 °C, during 15 min; MDA-EBC was measured by tandem liquid chromatography–mass spectrometry	Cr-EBC levels correlated with those of H2O2-EBC and MDA-EBC, as well as with urinary Cr levels.	14 Moderate
Casimirri, 2015, Italy [35]	Casimirri, 2015, Italy	Casimirri, 2015, Italy	Casimirri, 2015, Italy	Casimirri, 2015, Italy	Professional exposure to chlorinated agents increases EBC biomarkers of oxidativestress and inflammation	15 Moderate
Corradi, 2003, Italy [36]	To evaluate if aldehydes could be measured in EBC, to assess the influence of sampling procedures, to compare levels of different pulmonary disease groups with those of a control group	20 Patients with stable COPD (18 males and 2 females)	12 Smoking (9 males and 3 females) and 20 non-smoking (17 males and 3 females) control subjects	EBC was collected on a Tygon Tube immersed in thawing ice, and then frozen to −80 °C, no nose clip, MDA measured with LC-MS tandem	Aldehydes were identified in EBC, all, but were lower in control groups	16 Moderate
Corradi, 2002, Italy [37]	To investigate if short-term exposure to ozone (O3) induces changes in biomarkers of lung inflammation and oxidative stress in EBC		22 Non-smoking healthy control subjects (12 males and females)	EBC was collected on an ECO SCREEN device, during 15 min, frozen at −80 °C, MDA was measured as TBARS according to Nowak	A single 2-hour exposure to 0.1 ppm of O3 induces changes in biomarkers of inflammation and oxidative stress in those susceptible	11 Moderate
Corradi, 2004, Italy [38]	To compare aldehyde levels resulting from lipid peroxidation in EBC and induced sputum (IS) supernatant of subjects with asthma and chronic obstructive pulmonary disease	21 Subjects with COPD, 10 asthmatics	9 Healthy non-smoking control subjects (8 females and 1 male)	EBC was collected on a two glass chamber device from Incofar, no nose clip for 20 min, samples were stored at −80 °C, and MDA measured according to Larstad	Aldehydes can be detected in both exhaled breath condensate and supernatant of induced sputum, but their relative concentrations are different and not correlated with each other	12 Moderate
Doruk, 2011, Turkey [39]	To investigate oxidative stress in the lungs associated with tobacco smoke and to evaluate the effect of this stress with pulmonary function tests		69 Healthy subjects divided into 3 groups according to their exposure to tobacco smoke: 26 current smokers (23 males and 3 females), 21 subjects (15 males and 6 females) who did not smoke within the last year but had second-hand smoking and 22 (100 males and 12 females) with no tobacco smoke exposure	EBC was collected on an ECOSCREEN device for 15 min, wearing a nose clip, samples stored at −70 °C, MDA measured as TBARS	The levels of MDA, 8-OHdG, SOD, and GSH-Px were higher in smokers; NO levels gradually increased from Group I to Group III; MDA levels were lower in Group III than Group II	14 Moderate
Goldoni, 2004, Italy [40]	To investigate whether EBC can be used as a suitable matrix to assess target tissue dose and effects of inhaled cobalt and tungsten, using EBC malondialdehyde (MDA) as a biomarker of pulmonary oxidative stress	33 Workers exposed to Co and W in workshops producing either diamond tools or hard-metal mechanical parts	16 Control subjects (11 males and 5 females)	EBC was collected on a homemade apparatus, during 10 min, samples were transported in ice to the lab, and then stored at −80 °C, MDA was measured by liquid chromatography–tandem mass spectrometry (LC-MS/MS)	MDA levels were increased depending on cobalt concentration and were enhanced by coexposure to tungsten.	11 Moderate
Goldoni, 2013, Italy [41]	To compare the concentration of several biomarkers in whole (W-EBC) and fractionated EBC (A-EBC)		45 Healthy control subjects (10 males and 35 females, 6 smokers and 39 non-smokers)	EBC was collected on a homemade apparatus for the fractioned EBC and a TURBODECCS for the whole EBC for 15 min without a nose clip, EBC was centrifuged, and then stored at −80 °C, MDA was measured by liquid chromatography–tandem mass spectrometry	H2O2, 8-isoprostane, malondialdehyde, and 4-hydroxy-2-nonhenal were all higher in W-EBC, suggesting a contribution from the upper airways to oxidative stress biomarkers in apparently healthy subjects	10 Moderate
Goldoni, 2005, Italy [42]	To test the effect of condensation temperature on the parameters of exhaled breath condensate and the levels of selected biomarkers		24 Healthy control subjects (13 males and 11 females, 3 ex-smokers and 21 non-smokers)	EBC was collected on a TURBODECCS, during 10 min, at different temperatures, samples were centrifuged, and then stored at −80 °C, MDA was measured by liquid chromatography–tandem mass spectrometry (LC-MS/MS) within 2 weeks	Cooling temperature of exhaled breath condensate collection influenced selected biomarkers	13 Moderate
Gong, 2014, USA/China [43]	To compare ultrafine particles (UFPs) and fine particles (PM2.5) with respect to their associations with biomarkers reflecting multiple pathophysiological pathways linking exposure and cardiorespiratory events		125 Healthy non-smoking individuals (initially 64 males and 64 females) working on a campus near Peking’s University Health Sciences Centre	EBC was collected on an ECOSCREEN device, collected for 20 min, with nose clip, aliquotes stored at −70 °C, MDA was measured as MDA-TBA in HPLC	Associations of certain biomarkers with UFPs had different lag patterns as compared with those with PM2.5, suggesting that the ultrafine size fraction and the fine size fraction of PM2.5 are likely to affect PM-induced pathophysiological pathways independently	15 Moderate
Graczyk, 2016, Switzerland [44]	To investigate time course changes of particle-associated oxidative stress in exposed tungsten inert gas welders		20 Non-smoking healthy volunteers, with less than 1 year of apprentice in welding	EBC was collected on an Rtube, during 10 min, while wearing a nose clip, EBC samples were stored at −70 °C, MDA was measured by HPLC separation and fluorescence detection TBARS	A 60-minute exposure to TIG welding fume in a controlled, well-ventilated setting induced acute oxidative stress at 3 h post exposure in healthy, non-smoking apprentice welders not chronically exposed to welding fumes	10 Moderate
Gube, 2010, Germany [45]	To investigate the effect of welding as well as the impact of smoking and protection measures on biological markers in EBC	45 Male welders	24 Healthy males, non-exposed	EBC was collected on an ECOSCREEN device, wearing a nose clip, as long as 200 L of exhaled breath was collected, then stored at −80 °C, MDA was measued with a previously described but slightly changed method, i.e., MDA derivatized with DNPH and separated by HPLC, and then tandem mass spectrometry	Welders showed significantly increased concentrations of all these parameters at baseline as compared with non-exposed controls	14 Moderate
Inonu, 2012, Turkey [46]	To evaluate the differences in the burden of oxidative stress in patients with COPD, smokers, and non-smokers by measuring hydrogen peroxide (H_2_O_2_), malondialdehyde (MDA), and 8-isoprostane levels in EBC	25 Male COPD smokers	26 Smokers and 29 non-smokers, males, healthy	EBC collected on an EcoScreen device, 15 min with nose clip, and then quickly stored at −70 °C for 6 months, MDA was measured using a commercial kit in a fluorescence detector in HPLC	Even if respiratory function tests were within normal limits, oxidant burden in the lungs of smokers was equivalent to that in COPD patients, 8-isoprostane could be useful in assessing symptom severity and health status of COPD patients	16 Moderate
Larstad, 2001, Sweden [47]	To develop a method of MDA quantification	29 Patients with asthma, 7 of which with wheezing	15 control subjects without asthma	EBC was collected on an ECOSCREEN d evice, during 4 min, with a nose clip, stored at −20 °C until analysis, MDA was measured using TBS at 95 °C, and then separated by HPLC and MDA-TBA measured by fluorescence	No statistically significant difference between patients with asthma and patients without asthma; however, among females, subjects with asthma had higher MDA levels as compared with females without asthma; the use of the method when studying airway inflammation has to be further evaluated	12 Moderate
Laumbach, 2014, USA [48]	To determine if exposure to traffic-related pollutant particles (TRAPs) during commuting causes acute oxidative stress in the respiratory tract or changes in heart rate variability		21 Young volunteers (15 males and 6 females)	EBC was collected with an ECOSCREEN device, for 20 min, samples were frozen to −80 °C, MDA analysiswas performed by using a mixture of EBC, phosphoric acid, and thiobarbituric acid, which was heated and injected into an HPLC fluorescence system	Increases in markers of oxidative stress in EBC may represent early biological responses to widespread exposures to TRAPs particles that affect passengers in vehicles on heavily trafficked roadways	9 Low
Lee, 2015, Korea [49]	To investigate the actual health effects of multi wall carbon nanotubes in manufacturing workers	9 Male carbon nanotube manufacturing workers, 5 smokers	4 Office workers (3 males and 1 female), 2 smokers	EBC was collected on an Rtube, during 10 min, with a noseclip, stored in dry ice for 2 weeks. MDA was measured using fluorescence HPLC	Analyzed biomarkers in the MWCNT manufacturing workers were significantly higher than those in the office workers; MDA, n-hexanal, and molybdenum could be useful biomarkers of MWCNT exposure	10 Moderate
Majewska, 2004, Poland [50]	To determine whether concentrations of H_2_O_2_ and TBARs in EBC are elevated and correlate with systemic response to pneumonia during 10 days of hospital treatment	43 Inpatients with community acquired pneumonia (12 females and 31 males)	20 Healthy non-smoking control subjects (6 females and 14 males)	EBC was collected on a device from Jaeger, cooled with ethanol at −9 °C, during 20 min, wearing a nose clip, stored on ice until measurement was performed, MDA was measured as TBARS	Pneumonia is accompanied by oxidative stress in airways that moderately correlates with intensity of systemic inflammatory response, determination of H_2_O_2_ in EBC may be helpful for noninvasive monitoring of oxidant production during lower respiratory tract infection	13 Moderate
Nowak, 2001, Poland [51]	To investigate the concentration of H_2_O_2_ and TBARs in EBC and influencing factors		58 Healthy volunteers (18 smokers and 40 non-smokers, 31 males and 27 females)	EBC was collected on a device from Jaeger Toennies, during 20 min, wearing a noseclip, MDA was measured as TBARS spectrofotometrically	Neither moderate exercise nor one puff of salbutamol nor ipratropium significantly influenced the concentration of H_2_O_2_ and TBARs in EBC, only 4 of 120 EBC specimens from non-smoker subjects revealed detectable levels of TBARs, cigarette smokers exhaled more TBARs	11 Moderate
Pelclova, 2018, Czech Republic [52]	To investigate lung oxidative stress in workers handling nanocomposites	19 Nanocomposite-synthesizing and processing researchers (14 males and 5 females), all non-smokers	19 Control subjects (13 males and 6 females, all non-smokers)	EBC was collected on an ECOSCREEN device, until a minimum of 120 L of exhaled breath, wearing a nose clip, immediately stored at −80 °C, MDA was measured	Significant associations were found between working in nanocomposite synthesis and EBC biomarkers; more research is needed to understand the contribution of nanoparticles from nanocomposite processing in inducing oxidative stress, relative to other co-exposures generated during welding, smelting, and secondary oxidation processes	12 Moderate
Pelclova, 2014, Czech Republic [53]	To search for optimal markers in the exhaled breath condensate (EBC), plasma, and urine that would reflect the activity/severity of occupational asthma (OA) after withdrawal from the exposure to the allergen	43 Subjects with previously diagnosed immunological OA (18 males and 25 females)	20 Control subjects, working as office or healthcare employees and having no symptoms of asthma (10 males and 10 females)	EBC was collected for 15–20 min with EcoScreen, wearing a nose clip, samples were stored at −80 °C for a maximum of 2 months, MDA was measured with LC-ESI-MS/MS	Improvement in OA is very slow and objective impairments persist years after removal from the exposure; cysteinyl LTs and 8-ISO in EBC and 8-ISO in plasma might enrich the spectrum of useful objective tests for the follow-up of OA	12 Moderate
Pelclova, 2016, Czech Republic [54]	To investigate the utility of oxidative stress biomarkers in EBC in iron oxide pigment production workers	14 male workers, 43 ± 7 years, 43% smokers	14 Males, 39 ± 4 years, 50% smokers, non-exposed	EBC was collected for 15 min with an EcoScreen device, wearing a nose clip, samples were stored at −80 °C, MDA was measured with LC-ESI-MS/MS	Almost all markers of lipid, nucleic acid, and protein oxidation were elevated in the EBC of workers as compared with control subjects; markers in urine were not elevated	10 Moderate
Ricelli, 2018, Italy [55]	To investigate the cromium and nickel content of EBC of stainless steel (SS) tungsten inert gas (TIG) welders, and relate their concentrations with oxidative stress and inflammatory biomarkers		100 SS welding workers, aged 18–65 years, smokers/non-smokers/ex-smokers 33/36/31, values were considered pre-shift	EBC was collected on an ECOSCREEN device, during 15 min, and stored at −80 °C, MDA was determined by tandem LC-MS/MS	Given the weak relationship between the biomarkers and effects of exposure, we speculate that other substances generated during SS TIG welding also play a role in generating lung oxidative stress	11 Moderate
Rundell, 2008, USA [56]	To investigate PM1 inhalation during exercise on lung function, exhaled nitric oxide (eNO), total nitrate (NO3), S-nitrosoglutathione (GSNO), and malondialdehyde (MDA) in EBC		Twelve physically fit, non-asthmatic, non-smoking males	EBC was collected on an ECOSCREEN device, during 15 min, wearing a nose clip and stored at −80 °C, MDA was measured after derivatization with TBA by HPLC fluorescence	MDA increased 40% after low PM exercise, high PM1 inhalation during exercise caused a reduced alveolar contribution to eNO, NO3 and eNO variables were decreased and were related to impaired lung function	9 Low
Sakhvidi, 2015, Iran [57]	To measure exhaled breath malondialdehyde (EBC-MDA) in workers exposed to dust containing silica as compared with that of a non-exposed control group	25 Male workers in a ceramic factory	25 Male control subjects from administrative departments of the same factory	EBC was collected with a homemade apparatus, during 5 min, wearing a nose clip, samples were stored in a freezer at −20 °C until examination, MDA was analyzed using an HPLC equipped with a fluorescent detector	Significant correlation between respirable dust exposure intensity and the level of EBC-MDA of the exposed subjects, but no significant correlation with lung functions	13 Moderate
Sauvain, 2014, Switzerland [58]	To evaluate the feasibility of using exhaled breath condensate (EBC) from healthy volunteers for (1) assessing the lung deposited dose of combustion nanoparticles and (2) determining the resulting oxidative stress		15 Healthy non-smoking control subjects	EBC collected on a Rtube, during 10 min, wearing a nose clip, a aliquote was stored at −78 °C, MDA was derivatized with TBA, and then measured with HPLC fluorescence	The results suggest two phases of oxidation markers in EBC: first, the initial deposition of particles and gases in the lung lining liquid, and later the start of oxidative stress with associated cell membrane damage	11 Moderate
Stockfelt, 2012, Sweden [59]	To examine airway effects of two kinds of wood smoke in a chamber study		16 Healthy non-smoking control subjects (8 males and 8 females), 3 excluded for respiratory problems before the experiment	EBC collected according to Barregard, MDA measured always according to Barregard	Relatively low levels of wood smoke exposure induce effects on airways	8 Low
Syslova, 2009, Czech Republic [60]	To develop a sensitive method for a parallel, rapid and precise determination of the most prominent oxidative stress biomarkers for patients with silica or asbestos disease	20 Subjects with previous exposure to silica or asbestos and related disease	10 Control subjects	EBC was collected on an ECOSCREEN device, for 5−10 min, wearing a nose clip, frozen at −80 °C, and stored for a maximum of 1 month, MDA was measured with optimized LC–ESI-MS/MS	The differences in concentration levels of biomarkers between the two groups was perceptible in all the body fluids (the difference observed in an exhaled breath condensatewas statistically most significant)	12 Moderate
Szkudlarek, 2003, Poland [61]	To test whether exhalation of H_2_O_2_ and TBARs by healthy subjects depends on reactive oxygen species generation from blood phagocytes		41 Healthy, non-smoking control subjects, mean age 20.770.8 years (18 males and 23 females)	EBC was collected on a homemade mouth piece connected to a glass tube cooler, wearing a nose clip, during 20 min, MDA was measured as TBARS according to Nowak	No association between exhaled TBARs and blood phagocytes activity was found	10 Moderate
Pelclova, 2020, Czech Republic [62]	To analyze biomarkers in EBC of nanocomposite workers and understand the health effects	20 Researchers handling nanocomposites (15 males and 5 females), 19 non-smokers, mean age of 41.8 years	21 Control subjects (15 males and 6 females), office emplyees, 19 non-smokers, mean age of 42.7 years	EBC was collected on ECOSCREEN and stored at −80 °C, MDA was measured with LC-ESI- MS/MS	among inflammation markers, LT4 and tumor necrosis factor were the most useful	10 moderate

**Table 2 toxics-10-00258-t002:** Mixed effects regression analysis according to analytical method, use of nose clip, smoking status, and age category.

Parameters	β Coefficient	[95% Conf. Interval]	P > |t|
Analytical method			
HPLCfluo/TBA (n = 9)	reference		
Spectro/TBA (n = 2)	0.30	[−0.902; 1.49]	0.59
MS/DNPH (n = 9)	−1.03	[−2.02; 0.032]	0.04
HPLCfluo/NA (n = 2)	1.26	[−0.079; 2.6]	0.06
Factor-specific Wald test			0.04
Use of nose clip			
Yes (n = 9)	reference		
No (n = 8)	0.64	[−0.52; 1.81]	0.25
NA/other (n = 5)	1.14	[0.32; 1.96]	0.01
Factor-specific Wald test			0.04
Smoking habit			
Non-smokers (n = 11)	reference		
Mixed population (n = 3)	−0.71	[−1.70; 0.28]	0.14
Smokers (n = 5)	0.8	[0.08; 1.51]	0.03
NA (n = 3)	0.37	[−0.57; 1.31]	0.4
Factor-specific Wald test			0.04
Mean age (years)			
<30 (n = 4)	reference		
30–<40 (n = 8)	0.88	[−0.014; 1.77]	0.05
40–<50 (n = 3)	0.82	[−0.51; 2.16]	0.2
≥50 (n = 7)	1.21	[0.09; 2.33]	0.04
Factor-specific Wald test			0.15
Intercept	−1.15	[−2.07; −0.22]	0.02

HPLCfluo/TBA—high-performance liquid chromatography fluorescence/thiobarbituric acid; spectro/TBA—spectrophotometry/thiobarbituric acid; MS—mass spectroscopy; NA—information not available.

**Table 3 toxics-10-00258-t003:** Predicted concentrations of MDA (ng/mL) in EBC using the linear model.

Parameters	GM	95% Confidence Interval
Analytical method		
HPLCfluo/TBA (n = 9)	1.8	[0.795; 2.814]
Spectro/TBA (n = 2)	2.43	[0.142; 4.710]
MS/DNPH (n = 9)	0.65	[0.279; 1.015]
HPLCfluo/NA (n = 2)	6.35	[−0.252; 12.954]
Use of nose Clip		
NA/other (n = 5)	3.72	[1.150; 6.292]
No (n = 8)	2.27	[0.141; 4.403]
Yes (n = 9)	1.19	[0.656; 1.727]
Smoking habits		
Non-smokers (n = 11)	1.2	[0.751; 1.665]
Mixed population (n = 3)	0.6	[0.074; 1.111]
Smokers (n = 5)	2.69	[1.297; 4.065]
NA (n = 3)	1.75	[0.294; 3.207]
Mean Age (years)		
<30 (n = 4)	0.68	[0.186; 1.178]
30–<40 (n = 8)	1.64	[0.767; 2.520]
40–<50 (n = 3)	1.56	[0.238; 2.875]
≥50 (n = 7)	2.28	[1.150; 3.425]

## Data Availability

Not applicable.

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
