# Peer review of "Towards Reference Values for Malondialdehyde on Exhaled Breath Condensate: A Systematic Literature Review and Meta-Analysis"

_toxics, 2022, doi:10.3390/toxics10050258_

Round 1

Reviewer 1 Report

Review of the manuscript

Turcu V., Wild P., Hemmendinger M., Sauvain J.-J., Bergamaschi E., Hopf N.B., Canu I.G.:

Towards reference values for malondialdehyde on exhaled breath condensate: a systematic litearature review and meta-analysis

 submitted to Toxics (MDPI)

 General comment:

The manuscript addresses the use of malondialdehyde (MDA), an end-product of endogenous oxidation of lipids, as a biomarker of oxidative stress measured in exhaled breath condensate (EBC). The aim of the study was to define reference values for MDA in EBC by means of meta-analysis of relevant literature sources evaluated using a standardized quality assessment checklist (25 studies selected for final quantitative synthesis). Due to significant heterogeneity of data resulting from the above studies, possible influence of a set of confounding factors (age, gender, smoking status, EBC collection device, duration of EBC collection, use of nose clip, and analytical method) was evaluated using statistical methods. The effect of some of these factors was indeed confirmed. For this reason and also for methodological limitations encountered in the individual studies, the reference concentration interval for MDA in EBC couldn´t be determined. The study is presented clearly, study methods appear to be appropriate (the reviewer is not an expert in this field) and the conclusions are sound.

Detailed comments:

  • Major part of the manuscript is covered by extensive Table 1 consisting of characteristics of 40 studies, of which 25 was used for quantitative analysis. As the relevant data from these studies are presented in other tables and figures, I recommend placing Table 1 in Supplementary Materials. On the other hand, the current Supplementary Table S1 (Quality assessment criteria) should be presented in main body of the manuscript as a part of the Materials and Methods section.
  • Figure 2: I recommend to show the number of studies represented by each boxplot.
  • For a reader not familiar with Forest plots it would be useful to explain the meaning of brown rhomboidal marks presented in Fig. 3.
  • Though a generally valid reference value for MDA in EBC couldn´t be determined, the authors should still comment on suitability of this biomarker within individual studies where studied versus control groups are compared.  

Author Response

Dear Reviewer 

I agree with all your suggestion. 

I have implemented almost all of them, but concerning the number of studies in the box plots, I have to wait for an answer from the statisticien who performed the analysis, as I can not retrieve by myself this information.

Please find attached  the file modified accordingly. 

Yours, 

Veronica Turcu

Reviewer 2 Report

The study was well conducted and the paper is well written

 I only have one minor commnet.

In table1 the findings of the papaers of Antus 2013 and Antus 2015 are missing.

Author Response

Dear Reviwer, 

thank you for your attentive remark. It was a silly mistake on my behalf. 

I shall send the new version as currently I am waiting for a response from our statisticien to implement all received remarks. 

Yours, 

Veronica Turcu